# Thyroid-Bed Schwannoma Mimicking a Thyroid Neoplasm: A Challenging Diagnosis: Report of a Case and Literature Review

**DOI:** 10.3390/medicina58101345

**Published:** 2022-09-24

**Authors:** Claudio Gambardella, Ludovico Docimo, Giancarlo Candela, Giovanni Cozzolino, Federico Mongardini, Francesca Serilli, Giusiana Nesta, Marcello Filograna Pignatelli, Sonia Ferrandes, Antonio Gambardella, Giovanni Docimo

**Affiliations:** Department of Medical and Advanced Surgical Sciences, University of Campania “Luigi Vanvitelli”, 80131 Naples, Italy

**Keywords:** neck schwannoma, neck neoplasm, thyroid-bed lesion, thyroid mass, total thyroidectomy

## Abstract

Background: Schwannomas, also called neurinomas, are rare benign tumors of the neural cells that can develop from the sheaths of nervous structures of several districts, although the most frequent sites are the cranial nerves (25%–45%). Rarely, cases show neck schwannomas in the thyroid parenchyma, while the cases of thyroid-bed schwannomas mimicking a thyroid-gland lesions are anecdotal. Methods: We report the case of a 70-year-old man with a preoperative-imaging diagnosis of a thyroid neoplasm, confirmed as Thyr 4 by fine-needle cytology. Results: During surgery, an extra-thyroidal lesion was discovered, determined to be a neck schwannoma through definitive pathology. A literature review of cases of thyroid-bed-lesion schwannomas misinterpreted as thyroid neoplasms was carried out. Conclusions: In the case of suspicious extra-thyroidal lesions, we advocate for a close routine cooperation between the cytologist, the radiologist, and the surgeon in the attempt to reach an accurate preoperative diagnosis.

## 1. Introduction

Schwannomas, also called neurinomas, are rare benign tumors of the neural cells responsible for the myelin’s production. They can develop from the sheaths of the nervous structures of several districts, although the most frequent sites are branches of the cranial nerves, of the peroneal or ulnar nerve, or of the spinal roots [1]. They derive from the abnormal multiplication of the Schwann cells that compose the myelin sheath placed in the lining and isolation of the main branches of the peripheral nerves. Schwannomas usually affect people between the ages of 50 and 60. No sex or racial predilection is recognized. The incidence is 4.4–5.23 cases in 100,000 and accounts for 7% of primary tumors of the central nervous system (CNS) in adults [2,3]. Generally, it is an indolent pathology with only limited clinical manifestations such as monolateral tinnitus, dizziness, balance disorders, headache, feeling of pressure or fullness of the ear, otalgia, trigeminal neuralgia, and facial hypostenia for facial nerve involvement [4]. The localization and the characterization of the tumor mass are performed using a computed tomography (CT) scan or magnetic resonance imaging (MRI).

Between 25% and 45% of schwannomas originate from the head and neck nervous structures [5,6]. Rarely, cases of neck schwannomas have been described as thyroid-glandular parenchyma (primary-thyroid schwannomas) [7,8,9,10,11]. We herein present an anecdotal case of an extra-thyroidal neck schwannoma misinterpreted as a thyroid-gland tumor. Moreover, a systematic review of extra-thyroidal lesions mimicking a thyroid neoplasm is carried out.

## 2. Case Report

A 70-year-old man was referred to our surgical division for the presence of a palpable mass in the right neck region in November 2021. The patient complained of hoarseness and neck discomfort. We performed biochemical tests with the following results: FT3 4.57 pmol/L, FT4 17.50 pmol/L, TSH 1.38 microUI/mL, HGT 21.4 ng/mL, antiperoxidase antibodies 4.8 UI/mL, anti-thyroglobulin antibodies 11 UI/mL, and calcitonin 3.5 pg/mL. The patient underwent a neck ultrasound (US) that showed an enlarged thyroid gland, especially the right lobe (45 × 34 × 53.5 mm, left lobe 24 × 13 × 41 mm, isthmus diameter 3 mm). The glandular echoic structure was uneven. In the right lobe, the US showed a hypoechoic nodule of a maximum diameter of 47.5 mm with regular margins and with some contextual anechoic areola (Figure 1). Subsequently, we performed a computed tomography (CT) scan of the patient’s neck which confirmed the hyperplasia of the right lobe of the thyroid gland, almost completely occupied by a colliquate nodular formation (Figure 2). We performed a US-guided fine-needle cytology (FNC) of the latter lesion that highlighted, microscopically, the presence, in a hemorrhagic background, of a three-dimensional solid group of fused cells, immersed in a metachromatic matrix, in the absence of a colloid component (Figure 3).

The pathologists concluded that the mass was a Thyr 4 according to SIAPEC-AIT 2014 classification [12]. Therefore, considering the expected risk of malignancy (i.e., 60–80%), the patient was recommended to undergo a right hemithyroidectomy. In December 2021, during surgery, after the dissection of the prethyroidal muscles, the presence of an extra-thyroidal neoplasm was highlighted, easily dissociable from the gland. It was carefully dissected and removed. However, considering the presence of a suspicious nodularity and the preoperative diagnosis of a Thyr 4, the right thyroid lobe and the isthmus were also removed. The hospitalization was regular with an absence of complications of phonation or signs of hypocalcemia, and, after 2 days, the patient was discharged. Surprisingly, definitive pathology showed the presence of normal multinodular colloidal parenchyma of the right thyroid lobe and the isthmus. Furthermore, the extra-thyroidal neoplasm measured 5 × 3.5 cm and showed mesenchymal proliferation, spindle-cells type, with both dense and loose areas with thickened walls vases (Figure 4 and Figure 5). In dense areas (Antoni A), the so-called “Verocay bodies” were evident, while in the loosest areas (Antoni B), they appeared evident diffuse nuclear atypia (Figure 5). The Ki67 proliferation index showed a value lower than 5%; on immunohistochemistry, the protein S100 was positive while the relationship with actin was negative, concluding for a schwannoma (Figure 6). After 11 months, the patient was followed up through outpatient visits, and he did not show any sign of relapse of the pathology.

## 3. Review and Discussion

Schwannomas are benign tumors that can develop along the course of any nerve. For unknown reasons, the most frequent schwannomas are located on the acoustic nerve, followed by neurinomas of the facial and the trigeminal nerves [13]. Therefore, schwannomas of the head and neck region are common, while ones involving the thyroid gland are unusual [9]. Non-epithelial tumors of the thyroid gland (neurinoma, teratoma, hemangioma, lipoma, lymphoma, and leiomyoma) are extremely rare, and only a few reports exist in the literature [14,15]. In 1964, Delaney et al. was the first to describe a case of schwannomas arising in the thyroid gland [16]. Subsequently, only a few cases have been reported in the literature, often in adults in the age range of 20–57 years [17,18,19,20,21].

Even more rare, as in the reported case, are schwannomas of the thyroid bed. They may arise from the sympathetic chain, glossopharyngeal, vagus, accessory, recurrent laryngeal, or hypoglossal nerves [3,22,23,24,25,26,27,28,29]. Their diagnosis is challenging because of the extremely low incidence and the frequent possibility of misdiagnosis as intrathyroidal nodule on ultrasonography. Thyroid-bed schwannomas can appear as intraglandular lesions on a US [22,23,24,25,26,27,28,29] and on advanced imaging techniques such as CT scan and magnetic resonance imaging (MRI), that should be considered for the assessment of an enlarged thyroid nodule or mass (larger than 3 cm) and differentiation of a thyroid mass from an adjoining neck mass [22,23,24]. In a CT scan, the schwannoma appears as a well-circumscribed homogeneous mass of soft-tissue density, with increased density in the peripheral region of the tumor, after the injection of a contrast agent, due to neovascularization [4].

Histologically, a schwannoma is easily differentiable from a differentiated thyroid mass. It is composed, in fact, of an intimate mixture of spindle cells forming highly cellular Antoni A areas and less cellular, myxoid Antoni B areas. On cytological examination, a schwannoma is characterized by the presence of cells with slender wavy nuclei, fibrillary stroma, nuclear palisading, and the presence of Verocay bodies. The diagnosis can be also confirmed by the immunocytochemical positivity for markers for neural sheath tumors (i.e., S-100 protein and CD34). The differential diagnoses include mesenchymal lesions (i.e., leiomyoma, solitary fibrous tumor, and hemangiopericytoma) and epithelial tumors (i.e., medullary carcinoma, leiomyoma, solitary fibrous tumor, hemangiopericytoma, and epithelial tumors such as medullary carcinoma, thymoma, spindle epithelial tumor with thymus-like differentiation (SETTLE), and hyalinizing trabecular adenoma) which share the presence of spindle-cell lesions [28]. 

Many of these tumors are very rare entities, even if the spindle-cell variant of medullary-thyroid carcinoma may often be a cause of misdiagnosis in evaluating neural tumors of the thyroid. Therefore, careful research of features such as the presence of amyloid, chromatin pattern, and the presence of fine red granularity of the cytoplasm in Giemsa smears would help in the identification of medullary carcinomas.

Since previous reports did not distinguish between thyroid-bed schwannomas and primary-thyroid schwannomas, using the PubMed database, a systematic review of the current literature was carried out, up to May 2022. The MeSH (Medical Subject Headings) search terms used were “thyroid”, “schwannoma”, “thyroid bed mass”, “endocrine” and “neuroendocrine tumors”. We observed that the thyroid-bed schwannoma was an extremely rare neoplasm. The keywords “schwannoma”, “extra-thyroidal mass”, “extra-thyroidal schwannoma”, “thyroid”, “thyroid gland”, “neuroendocrine”, and “neuroendocrine tumor” were used for the research. Several combinations of the keywords and MeSH terms were utilized as shown: “Extra-thyroidal schwannoma” and “Schwannoma of the thyroid bed”. These various terms were substituted during the search. References of the more relevant articles were manually searched. The last research was concluded on 15 May 2022.

The final article was realized in accordance with the Preferred Reporting Items for Systematic Reviews and Meta-Analyses Guidelines [30]. Moreover, the eligible articles were selected according to the modified Newcastle–Ottawa scale in order to satisfy the requirements of the current review. The scale range is from 0 to 9. The studies included were those presenting a score of 6 or higher [31].

The following data were extracted from the included studies: first author, year of data collection, year of publication, gender and age of the patient, clinicopathological characteristics, and definitive pathology.

The inclusion criteria of the study comprised the report of patients with a proven histopathological diagnosis of an extra-thyroidal schwannoma mimicking a thyroidal mass. All studies that failed to fulfill the established inclusion criteria and those not in English were excluded.

To the best of our knowledge, to date, only 14 thyroid-bed schwannomas mimicking a malignant thyroid nodule have been reported in the literature (Table 1).

In 1997, Mikosch was the first to report a case of a cervical mass mimicking a thyroid nodule, and after noting the suspicious presence of a neurilemmoma at the FNC, the definitive pathology confirmed the diagnosis of an extra-thyroidal schwannoma with Antoni A structures [3]. We, in case of hypoechogenic nodules in connection with the thyroid gland, suggested it be considered that extra-thyroidal structures may cause hypoechogenic patterns. In 2008, Donatini et al. reported three cases of these kinds of lesions, often mistaken as thyroid nodules, and two received the definitive diagnosis of extra-thyroidal schwannomas [27].

Several authors reported how thyroid-bed schwannomas were often mistaken for a thyroid nodule [22,23,24,25,26,27,28,29]. This is not surprising since they share both clinical (painless, hard, elastic, mobile with swallowing) and sonographic (round or elongated, tendency to hypoechogenicity, possibly cystic aspect, thickened wall, abundant internal- and peripheral vascularization) characteristics.

Ultrasonographically, schwannomas are described as well-defined, round, or ovoid masses with a thickened wall and abundant internal and peripheral vascularization. They present echogenicity of varying degrees, according to the composition of Antoni cells and cystic changes, but they generally exhibit hypoechogenicity similar to or lower than that exhibited by muscle tissue [32,33,34].

Even considering cytological evaluation, schwannoma diagnosis is extremely challenging. FNA, in fact, is an easily available and accurate procedure for most head and neck masses; however, for the diagnosis of schwannomas, the sensitivity of FNA is poor, from 0% to 40%, with unsatisfactory specimen rates for FNA from 36% to 50% [35,36,37]. This limitation of FNA is due to the histologic characteristics of schwannomas, such as their dense interstitial components, hypocellular Antoni B areas, and frequent cystic degeneration [26]. For the thyroid-bed schwannomas reported in the literature, the results of FNA in over half of cases (8/14, 57.2%) were non-diagnostic, with only four other cases concluding for spindle-cell lesions [3,22,23,24,25,26,27]. Recently, core-needle biopsy (CNB) has been proposed as a complementary method for FNA, and some studies that compared it with FNA concluded that CNB provided a higher diagnostic accuracy with less invasive histological and immunohistochemical characterization of many head and neck tumors, especially schwannomas [33,38]. Noteworthy, when a sample labeled “from a thyroid nodule” contains few thyrocytes, the cytologist may be forgiven for considering it inadequate. It is here that the person who performed the biopsy must have an input: they will know whether the sample was representative of the lesion. 

Regarding the treatment, different from the intrathyroidal schwannomas, thyroid-bed schwannomas can be resected without the need for a thyroidectomy [24,25], even if in some cases the lesion can be not dissociable from the thyroidal tissue regarding a hemithyroidectomy or, rarely, a total thyroidectomy [39,40]. It is worth commenting that, in some cases, even during the surgical exploration, it is hard to distinguish between an extra-thyroidal mass (i.e., schwannoma) and an intrathyroidal lesion, leading to unnecessary thyroid surgery. In these cases, the use of a frozen section to drive the surgical strategy is highly requested [23,24,25,26].

In the reported case, the first instrumental investigation carried out, due to the presence of hoarseness and discomfort, a US of the neck region showed the presence of a nodule inside the thyroid parenchyma, a diagnosis that was then erroneously confirmed by a CT scan. Moreover, as in many others reported cases, the FNC of the lesion misguided the diagnosis. The FNC, in fact, concluded for a Thyr 4, according to the SIAPEC-AIT 2014 classification, leading to the recommendation of a total thyroidectomy for the presence of a thyroid mass highly suspicious for malignancy. During the neck surgical exploration, the presence of an extra-thyroidal neoplasm was highlighted, easily dissociable from the gland, that was removed along with the right thyroid lobe and the isthmus due to the presence of suspicious nodularity and the preoperative diagnosis of Thyr 4 that prompted an overtreatment. Definitive pathology concluded the presence of an extra-thyroidal schwannoma in a multinodular goiter. Moreover, according to the literature, schwannomas of the neck are generally discovered within the third or fourth decades. To the best of our knowledge, the current oldest patient presenting a thyroid-bed schwannoma reported in the literature is 70 years of age.

Therefore, in the case of suspicious extra-thyroidal lesions or thyroid-bed neoplasms, we advocate for a close routine cooperation between the cytologist, the radiologist, and the surgeon in the attempt to reach an accurate preoperative diagnosis. Moreover, in cases of inconclusive FNC cytology, or in the presence of few thyrocytes or cells of mesenchymal origin, a schwannoma of the neck should be considered and, furthermore, the surgeon should always ask for a frozen section before deciding to perform thyroid surgery.

## Figures and Tables

**Figure 1 medicina-58-01345-f001:**
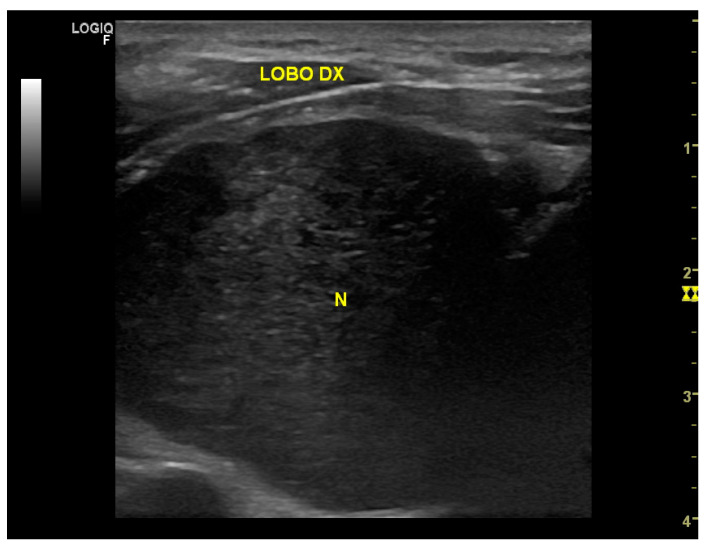
US image showing a hypoechoic nodule of a maximum diameter of 47.5 mm with regular margins and with some contextual anechoic areola.

**Figure 2 medicina-58-01345-f002:**
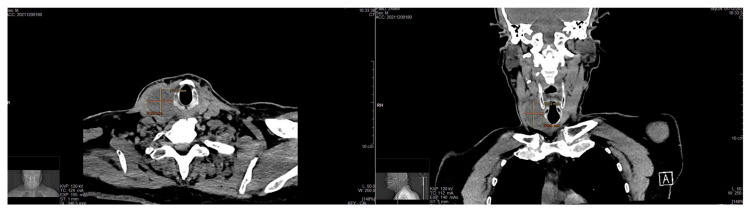
Axial and coronal CT scan images of the neck which confirmed the hyperplasia of the right lobe of the thyroid gland, almost completely occupied by a colliquate nodular formation.

**Figure 3 medicina-58-01345-f003:**
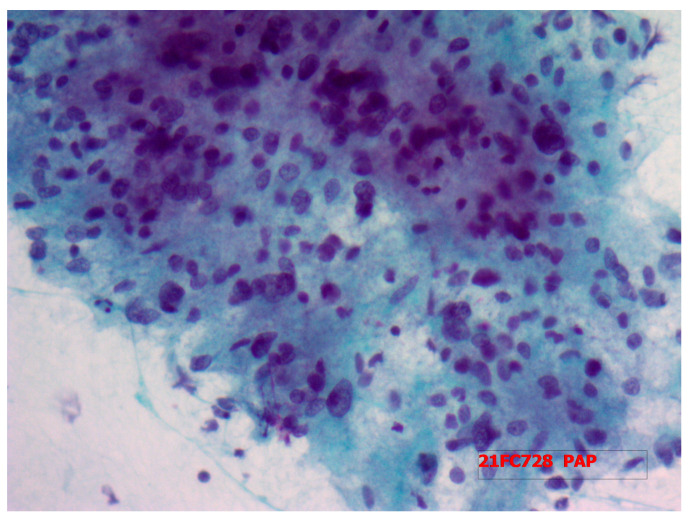
Fine-needle cytology of the neck lesion microscopically highlighting the presence of a three-dimensional solid group of fused cells in a hemorrhagic background, immersed in a metachromatic matrix, in the absence of a colloid component.

**Figure 4 medicina-58-01345-f004:**
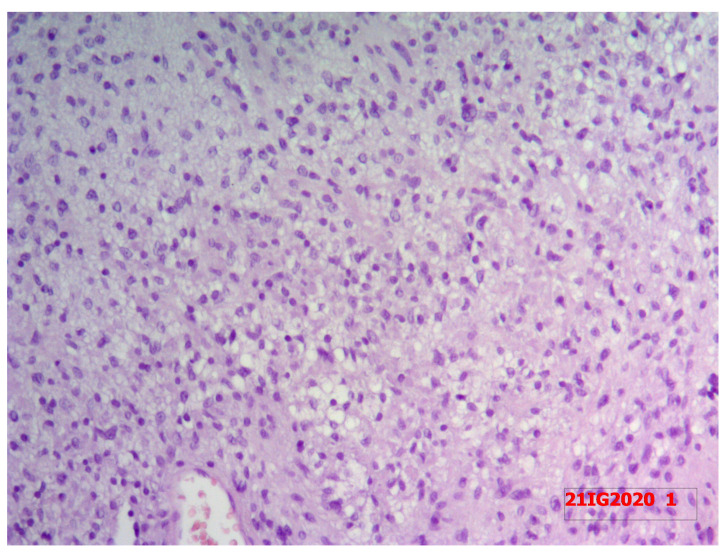
Resected specimen of extra-thyroidal neoplasm measuring 5 × 3.5 cm with the right thyroidal lobe and isthmus.

**Figure 5 medicina-58-01345-f005:**
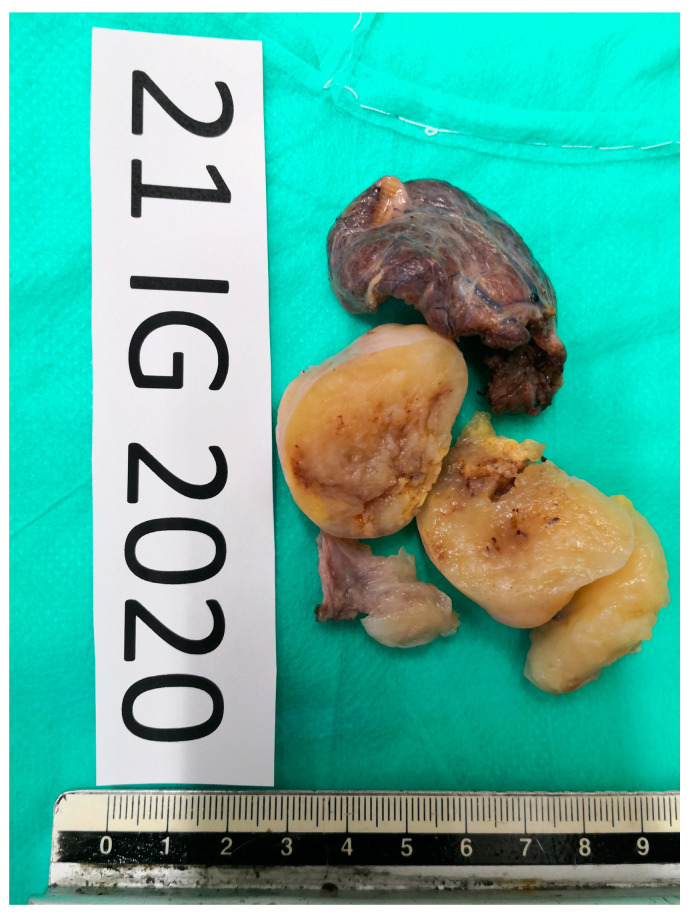
Definitive pathology showing mesenchymal proliferation, spindle-cells type, with both dense and loose areas with thickened walls vases at hematoxylin and eosin staining. In dense areas (Antoni A), the so-called “Verocay bodies” were evident, while in the loosest areas (Antoni B), they appeared evident diffuse nuclear atypia.

**Figure 6 medicina-58-01345-f006:**
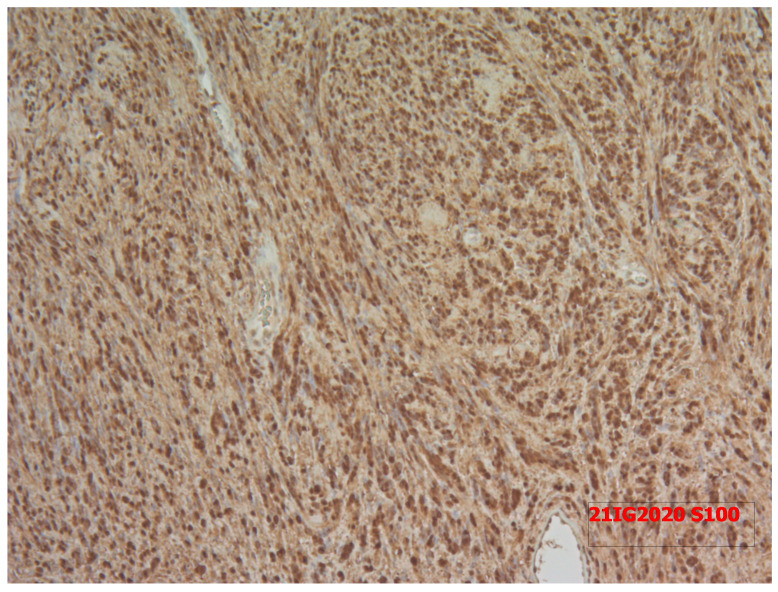
The Ki67 proliferation index showed a value lower than 5%; on immunohistochemistry, the protein S100 was positive while the relationship with actin was negative.

**Table 1 medicina-58-01345-t001:** Demographic, clinical, and histological features of patients affected by thyroid-bed schwannoma misinterpreted as thyroid neoplasm.

Case	Age/Sex	Presentation	US Localization	Imaging	FNA Result	Surgery	Definitive Pathology	IHC
Ledgard C et al., 2022 [25]	33/F	Large palpable mass, compression	53 × 19 × 19 mm—Inferior left thyroid nodule	US: Solid and hypoechoic with minimal internal vascularity (TIRADS 4)	Non-diagnostic (acellular)	Nodule Removal	Encapsulated mass (schwannoma) arising adjacent to nerve bundles. Spindle cells with both Type A and B Antoni cells	S-100 and Vimentin+ Ki67 low proliferation
Kang et al., 2019 [22]	33/F	None	30 mm—Inferior left thyroid nodule	US: Well-defined, oval-shaped, markedly hypoechoic intrathyroidal nodule with echogenic foci and macro- and microcalcifications	Non-diagnostic	Left hemi-thyroidectomy	Schwannoma in the perithyroid tissue, with compact areas of spindle cells (Antoni A) and loosely arranged foci (Antoni B)	S-100+
Nagavalli et al., 2017 [28]	60/F	Dysphagia, neck mass	Left thyroid lobe	US: Ovoid hypoechoic mass with smooth borders. CT: diffusely enlarged and heterogeneous thyroid with the left lobe extending to the retropharyngeal space and anterior mediastinum	Spindle cells with a lymphocytic background	Nodule Removal	Spindle cells in whorls with a predominantly Antoni A pattern	S-100+ Ki67 low proliferation
Simo D et al., 2014 [26]	-	Preoperative suspect of thyroid carcinoma with lymph node metastases	-	-	-	-	Ancient schwannoma	-
Pillai et al., 2013 [1]	30/F	Neck mass, voice change, dyspnea	5.3 × 4.1 × 7.2 cm—Lower pole of left thyroid lobe	US: Large multiseptated hypoechoic lesion. Not vascularized. CT: left thyroid mass measuring 8 × 5 cm with retrosternal extension. Tc-99 m scintigraphy: enlarged right lobe of thyroid and a left lateral palpable nodule, which was cold in nature, possibly an extra-thyroidal mass	Insufficient for diagnosis	Left hemi-thyroidectomy	Cellular ancient schwannoma with spindle cells with wavy nuclei, interspersed throughout hypocellular and cystic areas, extensive hyalinization, hemosiderin-laden macrophages and hyalinized thick-walled vessels.	S-100 and Vimentin+
Donatini et al., 2008/1 [27]	26/F	Neck mass	41 mm—Inferior right thyroid lobe	US: heterogeneous, partially liquid and with hyper-echogenic spots. Tc-99 m scintigraphy: no hyperactivity. MRI: extra-thyroidal lesion of 47 mm between the vascular neck bundle and the right thyroid lobe	Non-diagnostic	Mass Removal	Ganglioneuroma	S-100+
Donatini et al., 2008/2 [27]	26/F	Thyroid cancer with cervical-lymph-node metastases	38 mm—Right thyroid lobe	US: nodular lesion close to the neck vascular bundle suspicious for a metastatic lymph node. CT and MRI: enlarged left thyroid lobe pushing the trachea towards right side and in close proximity to a largely necrotic lesion of 4 cm (metastatic disease). Tc-99 m scintigraphy: the “lymph node” showed hypo-activity	Fibrous tissue without any tumoral cell	Mass Removal	Schwannoma	S-100+ Ki67/Mib1-
Cashman et al., 2008 [32]	35/F	Neck mass; right Horner’s syndrome	6 × 2.4 × 2.7 mm—Right thyroid lobe	US: Large mass arising from thyroid gland	Inconclusive	Right hemi-thyroidectomy	Schwannoma with spindle cell without atypia, mitosis, or necrosis	S-100+ Desmin and SMA (smooth muscle actin)
De Paoli et al., 2005 [23]	63/F	Foreign body sensation with swallowing	27 mm—Lower pole of right thyroid lobe	US: Markedly hypoechogenic nodule with rich vascularity	Fragments of adipose tissue, rare thymocytes in aggregates resembling follicular masses, insufficient for diagnosis	Total thyroidectomy	Schwannoma in the perithyroid tissue, with compact areas of spindle cells or Verocay bodies (Antoni A) and loosely arranged foci (Antoni B)	
Aron et al., 2005/1 [13]	28/F	Neck mass	Left lobe	-	Spindle-cell tumor without thyroid cell. Diagnosis of benign nerve-sheath tumor	Left hemi-thyroidectomy	Grey-white lobulated tumor of 70 × 50 mm, with compressed normal thyroid tissue at the periphery. Microscopic examination showed features of a schwannoma with nuclear palisading and thick-walled blood vessels	-
Aron et al., 2005/2 [13]	23/F	Neck mass	-	-	Spindle-cell tumor without thyroid cell. Diagnosis of benign nerve-sheath tumor	Total thyroidectomy	Grey-white lobulated tumor of 80 × 50 mm, with compressed normal thyroid tissue at the periphery. Microscopic examination showed features of a schwannoma with nuclear palisading and thick-walled blood vessels	-
Badawi et al., 2002 [29]	23/F	Neck mass	-	Tc-99 m scintigraphy: cold nodule of the left thyroid lobule	Paucicellular sample with occasional follicular cells of equivocal diagnostic value	Mass excision	Ancient, encapsulated schwannoma with vague nodularity with fibrous septae. Lobules consisted of collagenous stroma and cells with Verocay bodies.	-
Ahmed et al., 2000 [6]	14/M	Neck mass	Superior pole to right thyroid lobe	Solid lesion	-	Mass excision	Neurilemmoma	-
Mikosch et al., 1997 [3]	31/M	Neck mass	Lateral and lower pole of right thyroid lobe	US: Large markedly hypoechoic nodule with smooth borders. Tc-99 m scintigraphy: normal thyroid	1. Colloid and thyrocytes 2. Spindle-cell tumor	Mass excision	Schwannoma with Antoni A structures	-

## Data Availability

The datasets used and/or analyzed during the current study are available from the Unit of Thyroid Surgery, University of Campania “Luigi Vanvitelli”, Napoli, Italy, upon reasonable request.

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
