# Peer review of "Thyroid-Bed Schwannoma Mimicking a Thyroid Neoplasm: A Challenging Diagnosis: Report of a Case and Literature Review"

_medicina, 2022, doi:10.3390/medicina58101345_

Round 1

Reviewer 1 Report

dear authors, 

The subject of your manuscript in very interesting, the tumor type ans the localization being very rare.

In the case report I think it would be useful to show CT aspect and their explanation as well as particularities of the surgery and the surgical technique chosen for the ablation of the schwanomma.

I think it would be useful to describe the status of the patient after surgery.

In the ,,Discussions,, I Think you should emphasize the particularities of your case compared to those reported in the literature.

The conclusion of the study must be also mentioned.

Best regards!

Author Response

Thank you for your precious comments and suggestions.

A Ct scan image has been added and the surgical technique has been more detailed. Moreover the particularity of the case has been detailed in the discussion. 

Reviewer 2 Report

An interesting manuscript on atypical sites of schwannoma.

It seems original and well structured, but some aspects have to be addressed.

 In the case presentation section, it would be interesting if the authors would provide some Computed Tomography images. In the same section, the authors did not provide any information about the postoperative evolution.

Best regards and good luck.

Author Response

Thank you for your precious comments and suggestions.

A Ct scan image has been added and information about the postoperative course have been provided

Reviewer 3 Report

What did the ultrasound report say in this case were there not able to diagnoses a different sonological appearance .Was the FNAC ultasound guided?

in how many of your references were they able to diagnose an extrathyroidal lesion on ultra sound 

i think an editing of the article needs tdo be done

Author Response

Thank you for the opportunity to clarify. The report of the US is detailed in the case report section as follows: "The patient underwent to neck Ultrasound (US) that showed an enlarged thyroid gland, especially the right lobe (45x34x53.5mm, left lobe 24x13x41 mm, isthmus diameter 3 mm). The glandular echoic structure was uneven. In the right lobe, US showed a hypoechoic nodule of maximum diameter of 47.5 mm with regular margins and with some contextual anechoic areola" It was performed by an external operator and I was not able to say why he does not recognize the lesion. Moreover, The FNC was US guided. This was added in the text

Regarding the second point, as reported in the table, onli 1 case out of 14 the physicians were able to diagnose an extrathyroidal lesion on simply ultrasound 

Round 2

Reviewer 3 Report

corrections made can be accepted